# Factors associated with recovery from stunting at 24 months of age among infants and young children enrolled in the Pediatric Development Clinic (PDC): A retrospective cohort study in rural Rwanda

Mathieu Nemerimana[1]*, Silas Havugarurema[2], Alphonse Nshimyiryo[1], Angelique Charlie Karambizi[1], Catherine M. Kirk[1], Kathryn Beck[1], Chantal Gégout[3], Todd Anderson[1], Olivier Bigirumwami[4], Jules Maurice Ubarijoro[1], Patient K. Ngamije[2], Ann C. Miller[5]

1 Partners In Health/Inshuti Mu Buzima, Kigali, Rwanda, 2 Kirehe District Hospital, Kirehe, Rwanda, 3 World Health Organization, Kigali, Rwanda, 4 Rwinkwavu District Hospital, Kayonza, Rwanda, 5 Department of Global Health and Social Medicine, Harvard Medical School, Boston, Massachusetts, United States of America

* mnemerimana@pih.org

## Abstract

### Introduction

Stunting (low height/length-for-age) in early life is associated with poor long-term health and developmental outcomes. Nutrition interventions provided during the first 1,000 days of life can result in improved catch-up growth and development outcomes. We assessed factors associated with stunting recovery at 24 months of age among infants and young Children enrolled in Pediatric Development Clinics (PDC) who were stunted at 11 months of age.

### Methods

This retrospective cohort study included infants and young children who enrolled in PDCs in two rural districts in Rwanda between April 2014 and December 2018. Children were included in the study if their PDC enrollment happened within 2 months after birth, were stunted at 11 months of age (considered as baseline) and had a stunting status measured and analyzed at 24 months of age. We defined moderate stunting as length-for-age z-score (LAZ) < -2 and ≥-3 and severe stunting as LAZ <-3 based on the 2006 WHO child growth standards. Stunting recovery at 24 months of age was defined as the child's LAZ changing from <-2 to > -2. We used logistic regression analysis to investigate factors associated with stunting recovery. The factors analyzed included child and mother's socio-demographic and clinical characteristics.

### Results

Of the 179 children who were eligible for this study, 100 (55.9%) were severely stunted at age 11 months. At 24 months of age, 37 (20.7%) children recovered from stunting, while 21 (21.0%) severely stunted children improved to moderate stunting and 20 (25.3%)

**Data Availability Statement:** All relevant data are within the paper.

**Funding:** During the study period, the PDC program has been supported with funding through Grand Challenges Canada Saving Brains (R-SB-POC-1707-09583), the Primates World Relief and Development Fund via Global Affairs Canada (D-001975-001) and Partners In Health/Inshuti Mu Buzima. There was no additional external funding received for this study. The funders had no role in study design, data collection and analysis, decision to publish, or preparation of the manuscript.

**Competing interests:** The authors have declared that no competing interests exist.

moderately-stunted children worsened to severe stunting. Early stunting at 6 months of age was associated with lower odds of stunting recovery, with the odds of stunting recovery being reduced by 80% (aOR: 0.2; 95%CI: 0.07–0.81) for severely stunted children and by 60% (aOR: 0.4; 95% CI: 0.16–0.97) for moderately stunted children (p = 0.035). Lower odds of stunting recovery were also observed among children who were severely stunted at 11 months of age (aOR: 0.3; 95% CI: 0.1–0.6, p = 0.004). No other maternal or child factors were statistically significantly associated with recovery from stunting at 24 months in our final adjusted model.

## Conclusion

A substantial proportion of children who were enrolled in PDC within 2 months after birth and were stunted at 11 months of age recovered from stunting at 24 months of age. Children who were severely stunted at 11 months of age (baseline) and those who were stunted at 6 months of age were less likely to recover from stunting at 24 months of age compared to those with moderate stunting at 11 months and no stunting at 6 months of age, respectively. More focus on prevention and early identification of stunting during pregnancy and early life is important to the healthy growth of a child.

## Introduction

Stunting (low height- or length-for-age) remains a major public health problem with about 149 million children under five years affected globally; 39% of these children are from Africa [1]. Stunting in early life is associated with long-term challenges including increased morbidity, mortality, chronic diseases and infections, inadequate physical functioning, low economic productivity, and poor cognition and neurodevelopment [2–4]. Children with persistent childhood stunting experience long-term poor developmental outcomes [5, 6]. Small size at birth is one of the leading contributors to stunting globally [7]. In Rwanda, stunting rates in under-five children are higher compared to global rates, 38% [8] versus 25% [1]. The situation is even worse for children born small or who become sick in the first weeks of life. "Small and sick" children (defined by the WHO as a child weighing <2500g at birth or a newborn with any medical or surgical condition [9]) have nearly double the rates of stunting compared to national averages [8, 10]. About 70% of the burden of stunting by 59 months of age is rooted from the first 24 months of life [11] and the rates of stunting increase as children age, peaking at 23 months; after this age the rates of newly stunted children tend to decrease [12].

The first 1,000 days of life, from conception to two years, is a critical period for child development when the brain is most rapidly developing and is highly sensitive to program interventions for improving child growth and developmental outcomes [7, 13]. A study of countries that have had outsized success at reducing stunting relative to economic growth has identified several interventions, particularly during pregnancy, that have contributed to stunting declines [14], but less is understood about successful interventions in early childhood. There is evidence that with sustained nutritional interventions throughout childhood until adolescence [15] infants who are stunted can have growth catch-up and recover from stunting, though recovery is thought to be most likely before age two [16]). Studies in Kenya, Brazil and South Africa have found varying rates of stunting recovery ranging from 19% to 45% [17–19]. In the South African study, factors associated with recovery from stunting included early child growth failure, birth spacing, monitoring and supporting growth, education status of the mother, birth

weight of the child and household socioeconomic factors [19]. Though factors associated with recovery from stunting among general child populations have been identified [17–19], little is known about recovery from stunting among high-risk children in low- and middle-income countries.

In 2014, Partners In Health/Inshuti Mu Buzima (PIH/IMB), in partnership with the Rwanda Ministry of Health and United Nations Children's Fund (UNICEF), initiated Pediatric Development Clinics (PDC) as a new integrated model of outpatient care at public district hospitals and health centers in rural settings to help all infants who are born small and sick survive, thrive and reach their full developmental potential. Through structured follow-up care and support, children in PDC receive integrated prevention, early detection and support interventions to improve their nutrition, health and development [20, 21]. This study aimed to examine, among children enrolled in PDCs in Rural Rwanda and stunted at 11 months, what proportion recovered from stunting at 24 months of age and which factors were associated with their recovery.

## Methods

### Study setting

The study was conducted at 10 PDCs located in the Rwinkwavu District Hospital (RDH) and Kirehe District Hospital (KDH) catchment areas. KDH and RDH are respectively located in Kirehe and Kayonza Districts in the rural Eastern Province of Rwanda. RDH and KDH are public hospitals that supervise 8 health centers and 19 health centers respectively. These health facilities serve about 600,000 people [22] and have been supported by PIH/IMB since 2005. The first PDC was initiated at RDH in 2014 and then expanded to four health centers by the end of 2015. The PDC was expanded to KDH in 2016 and four more health centers affiliated to KDH were opened between 2018–2019. By the time of data collection in 2020, PDC was being implemented at 10 health facilities in Rwanda and all were included in the study.

### Description of PDC program intervention

The criteria for referral to and enrollment in PDC include being born preterm, low birth weight (LBW) under 2,000 grams, hypoxic ischemic encephalopathy (HIE), cleft lip/palate, hydrocephalus, trisomy 21, global developmental delay, and infants aged less than 12 months following hospitalization for severe acute malnutrition. Eligible children are referred from neonatal care units or other hospital services or self-referral from community. Children enrolled in PDC are followed up with standard visits every week within first month of age, then at 1, 2, 4, 6, 9, and 12 months of age then every 6 months. Children are discharged at 36 months if they are developmentally on-track with good health and nutritional status while these with any developmental, health or nutritional challenges are continued to be followed up until they reach 5 years of age. At each PDC visit, children are screened for danger signs and provided clinical check-ups with linkages to specialized care for identified health issues or complications, and provided with nutrition packages, and family-centered developmental care for children with developmental difficulties, including parent coaching in nurturing care, counselling, food packages for nutritionally at-risk infants, and home visit for complicated need. More detailed descriptions of the PDC program intervention and protocol have been previously published [20, 21].

**Study population.** This study included children who were enrolled in PDCs from April 2014 to December 2018. Children were included only if they were enrolled within 2 months of birth and they were identified as stunted (length-for-age z-score (LAZ) <-2) at 11 (+/- 1)

months of age. We selected 11 months of age as the cutoff for inclusion to ensure that we included children whose growth faltered after transitioning from exclusive breastfeeding and to ensure the opportunity of a full year of access to PDC services prior to assessment. All eligible children were followed until they reached 24 months of age (using corrected age for children born preterm with a known gestational age) and continued to be enrolled in PDC. Corrected age was used for preterm infants for appropriate growth monitoring interpretations using WHO standards growth charts [23]. The study excluded children who did not have a PDC visit within 2 months of when they turned age 24 months or those who did not have a documented length at 24 months of age.

**Data collection.**  This was a retrospective cohort study, which assessed factors associated with stunting recovery at 24 months of age among infants and young children enrolled in the PDC within 2 months after birth who were stunted at 11 months of age.

We extracted data of children enrolled in PDC between April 2014 and December 2018 from the Electronic Medical Record (EMR) system; an open medical record system using OpenMRS that is used to electronically record client information in PDCs. All data on child and mother's socio-demographic, and clinical characteristics, including anthropometric measurements of children enrolled in PDC, are routinely recorded on standardized clinical charts by PDC care providers and later entered in the EMR by trained EMR data officers or trained PDC providers within 1 week of each PDC visit. Data on socio-demographic, clinical characteristics and anthropometric measurements of children meeting inclusion criteria were extracted from the EMR. We also extracted socio-demographic data of the mothers recorded in EMR. Socio-demographic data included age, sex, mother's marital status and number of years in school, household socioeconomic category (Ubudehe), and the total number of children in the household. We assessed the extracted data for completeness. Any missing data or data with identified errors were double checked and corrected when possible. The EMR team at PIH/ IMB has standard procedures for routine data quality assessments of EMR data. Data were de-identified by the EMR team prior to analysis.

**Definitions.**  PDCs were defined into two categories, hospital-based PDCs and health center-based PDCs. "Hospital-based PDCs" are described as PDCs located at the district hospital level. Both types of PDCs provide services following the same clinical protocols and tools; eligible children are enrolled based on the PDCs nearest to their home.

Our primary outcome was recovery from stunting at 24 months of age. We defined stunting as Length for Age Z score (LAZ) <-2 standard deviations below the mean based on the 2006 WHO Child Growth Standards [24]. "Moderate" stunting was defined as a z-score of <-2 and ≥-3 and "severe" defined as a z-score <-3 SDs. Unless specified otherwise, we categorized "stunting" as children with a z-score <-2 or ≥-2. Wasting was defined as weight-for-length z-score (WLZ) <-2. Underweight was defined as weight-for-age z-score (WAZ) <-2. Recovery from stunting at 24 months of age was defined as having a LAZ > -2 on the latest PDC visit closest to the child's age of 24 months (visit had to have occurred within +/- 2 months) using corrected age if the child was born with a documented gestational age <37 weeks.

Ubudehe is a system used by the government of Rwanda to categorize the social and economic status of households at a community level. There are four categories: category 1 (i.e. families who are very poor and vulnerable, not owning shelter and unable to feed themselves without support), category 2 (i.e. families with access to minimal owned or rented housing but no employment and food insecurity, e.g. only can afford to eat once or twice a day), category 3 (i.e. families who are employed; farmers with production beyond subsistence farming or are employers of labor; families who own small and medium enterprises), and category 4 (i.e. families who own large businesses, industries or companies; hold full-time employment by

organizations; or are public servants) [25]. For this study, we combined Ubudehe category 3 and 4 as they are not considered as poor and there were few children in category 4.

Corrected age was calculated for known instances of children born preterm by taking chronological age minus the weeks born preterm, where weeks born preterm is defined as 40 weeks minus gestational age. Chronological age was calculated by taking the date of each PDC visit minus the date of birth. We assessed gestational age (full term defined as 37+ weeks and preterm as <37 weeks completed gestation) and birthweight (normal weight: ≥2500 grams, LBW: 2000–2499 grams, moderate LBW: 1500–1999 grams, and very LBW: <1500 grams) as categorical variables. Small for gestational age (SGA) was defined as birth weight lower than the 10th percentile for gestational age based on INTERGROWTH-21st standards [26].

"Child diagnosed with other conditions" was defined as having any diagnosis other than preterm, low birth weight, or HIE. Other conditions were combined into separate variable due to the small number of children presenting with these conditions. These other conditions include central nervous system infections, trisomy 21, post-hospitalization for severe malnutrition when < 12 months of age, hydrocephalus, cleft lip or palate and other developmental delays. The variable "Child diagnosed with multiple conditions" was defined as children who were diagnosed with more than one condition. We included a category for missing data in most categorical variables.

**Data analysis.** We described the child and mother's demographic and clinical characteristics using frequencies and percentages for categorical data and median and interquartile ranges (IQR) for continuous variables. We reported the proportion of children who recovered from stunting at the PDC visit closest to 24 months of age. We used logistic regression to investigate socio-demographic and clinical factors associated with recovery from stunting at 24 months of age in bivariate analyses. We considered factors associated with the outcome at an $\alpha<0.20$ in the bivariate analysis for inclusion in the final model. A final reduced model was built using backward stepwise logistic regression; only factors showing associations at a $\alpha<0.05$ significance level were retained in the final model. However, because sex has been reported to be associated with differences in stunting among under five children [27], we left child sex in the final model to adjust for its effects. We excluded health insurance status in the final model due to a large proportion of missing values. We analyzed data using Stata v.15.1 (Stata Corp, College Station, TX, USA).

**Ethical considerations.** This study was reviewed and approved by the Partners In Health/ Inshuti Mu Buzima research committee and was conducted under the Pediatric Development Clinic (PDC) evaluation protocol as it was approved by the Rwanda National Ethics Committee (RNEC) (Approval Number: 713/RNEC/2019). Since this study relied on retrospective de-identified data, the requirement for written informed consent was waived.

## Inclusivity in global research

Additional information regarding the ethical, cultural, and scientific considerations specific to inclusivity in global research is included in the S1 Checklist.

## Results

In total, of 403 infants and young children who had documented stunting at 11 months, 179 met further eligibility criteria and were included in this study (Fig 1). Except the fact that ineligible kids were significantly more likely from PDCs at HCs, there was no other statistically significant differences existed between the included and excluded participants on baseline demographic or health status variables (S1 Table). Table 1 presents socio-demographic data of the study population.

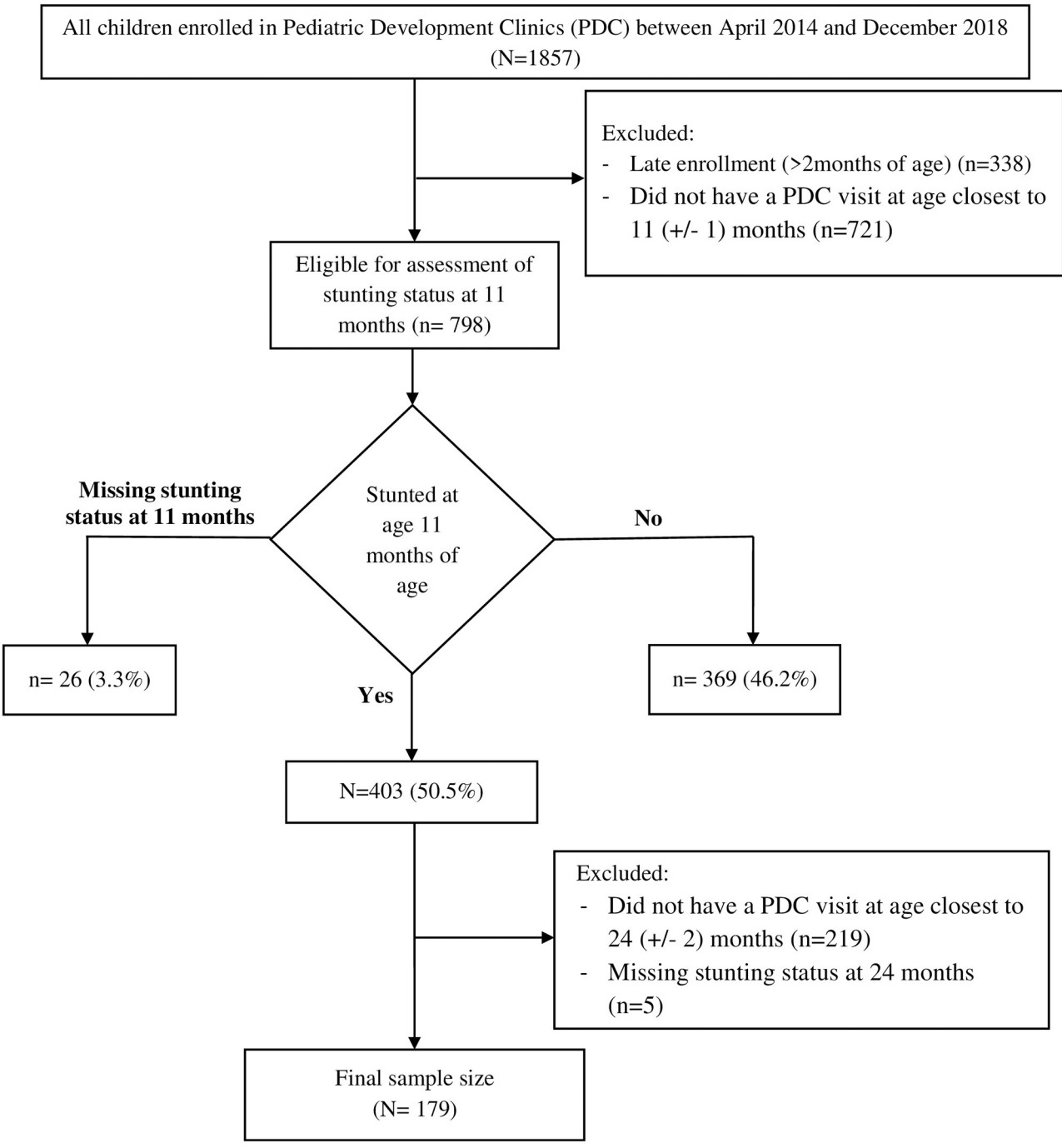

**Fig 1. Study flow chart.** This flow diagram illustrates the process for selection and enrollment of study participants.

Implementation of PDC was initially started in Kayonza district then expanded in Kirehe district, thus of the 179 infants and young children, the majority were from Kayonza district (73.7%). Our study population consisted of more males (60.9%), people with low household resources (Ubudehe category 1 or 2 (62.6%). At the time of enrollment in PDC, 7.8% of children did not have any health insurance and 10.6% of children were cared for by one parent

**Table 1. Socio-demographics of children stunted at 11 months and their mothers enrolled in PDC, N = 179 unless otherwise indicated.**

| | n | % |
|---|---|---|
| **District** | | |
| Kirehe | 47 | 26.3 |
| Kayonza | 132 | 73.7 |
| **PDC site** | | |
| Hospital PDCs | 102 | 57.0 |
| Health Center PDCs | 77 | 43.0 |
| **Child's age at the PDC visit closest to 24 months of age, median (IQR)** | 23.9 | (22.2, 25.9) |
| **Child's sex** | | |
| Male | 109 | 60.9 |
| Female | 70 | 39.1 |
| **Health insurance status** | | |
| No | 14 | 7.8 |
| Yes | 142 | 79.3 |
| Missing data | 23 | 12.8 |
| **Age of the mother at child's enrollment to PDC, median (IQR)** | 28 | (18, 43) |
| **Mother's marital status at child's enrollment in PDC** | | |
| Married | 97 | 54.2 |
| Cohabitating (Living with partner) | 49 | 27.4 |
| Single, divorced, of widowed | 19 | 10.6 |
| Missing data | 14 | 7.8 |
| **Caregiver's number of years in school, median (IQR)** | 5 | (0,10) |
| **Household socioeconomic category (Ubudehe)** | | |
| Category 1 (very poor) | 15 | 8.4 |
| Category 2 (poor) | 97 | 54.2 |
| Category 3 or 4 (not poor) | 44 | 24.6 |
| Unknown | 23 | 12.8 |
| **Total number of children in the household** | | |
| < 3 children | 70 | 39.1 |
| ≥ 3 children | 75 | 41.9 |
| Missing data | 34 | 19.0 |

Abbreviations: IQR, interquartile range; PDC, Pediatric Development Clinic

(mothers who are single, divorced or widowed). The median number of years of maternal education was 5 (IQR: 0, 10).

Table 2 presents clinical characteristics of the children in the study. Among the children enrolled in PDC with stunting at 11 months, most (77.6%) had a history of prematurity or LBW; smaller numbers of children were referred to PDC with other conditions.

Most children had a history of malnutrition at 6 months of age: (21.2%) infants had wasting, 67.0% were underweight and 73.7% were stunted. At 11 months of age, the percentages of infants with wasting and underweight was similar or slightly increased compared to 6 months. The median number of PDC visits between 11 and 24 months was 5 visits (IQR: 4,7).

As presented in Table 3, 37 (20.7%) recovered from stunting at 24 months of age, while 21 (21.0%) children with severe stunting improved to moderate stunting. However, 20 (25.3%) worsened from moderate to severe stunting.

**Table 2. Clinical characteristics of children stunted at 11 months and their mothers enrolled in PDC, N = 179 unless otherwise indicated.**

|  | n | % |
|---|---|---|
| **Gestational age** | | |
| ≥37 weeks | 46 | 25.7 |
| <37 weeks | 81 | 45.2 |
| Missing data | 52 | 29.0 |
| **Child's weight at birth** | | |
| ≥2500g | 31 | 17.3 |
| 2000-2499g | 24 | 13.4 |
| 1500-1999g | 72 | 40.2 |
| <1500g | 37 | 20.7 |
| Missing data | 15 | 8.4 |
| **Small for gestational age** | | |
| No | 43 | 24.0 |
| Yes | 76 | 42.5 |
| Missing data | 60 | 33.5 |
| **Child diagnosed as preterm or LBW** | | |
| No | 37 | 20.7 |
| Yes | 139 | 77.6 |
| Missing | 3 | 1.7 |
| **Child diagnosed with HIE** | | |
| No | 137 | 76.5 |
| Yes | 39 | 21.8 |
| Missing | 3 | 1.7 |
| **Child diagnosed with other conditions[a]** | | |
| No | 169 | 94.4 |
| Yes | 7 | 3.9 |
| Missing | 3 | 1.7 |
| **Child diagnosed with multiple conditions[b]** | | |
| No | 167 | 93.3 |
| Yes | 9 | 5.0 |
| Missing | 3 | 1.7 |
| **Wasting status at intake visit** | | |
| Not wasted (Normal WLZ) | 83 | 46.4 |
| Moderate wasting | 40 | 22.3 |
| Severe wasting | 17 | 9.5 |
| Missing data | 39 | 21.8 |
| **Underweight status at intake visit** | | |
| No underweight (Normal WAZ) | 26 | 14.5 |
| Moderate underweight | 22 | 12.3 |
| Severe underweight | 66 | 36.9 |
| Missing data | 65 | 36.3 |
| **Wasting status at closest visit to 6 months of age** | | |
| Not wasted (Normal WLZ) | 137 | 76.5 |
| Moderate wasting | 26 | 14.5 |
| Severe wasting | 12 | 6.7 |
| Missing data | 4 | 2.2 |
| **Underweight status at closest visit to 6 months of age** | | |

(*Continued*)

**Table 2.** (Continued)

| | n | % |
|---|---|---|
| No underweight (Normal WAZ) | 57 | 31.8 |
| Moderate underweight | 55 | 30.7 |
| Severe underweight | 65 | 36.3 |
| Missing data | 2 | 1.1 |
| **Stunting status at closest visit to 6 months of age** | | |
| Not stunted (Normal LAZ) | 42 | 23.5 |
| Moderate stunting | 71 | 39.7 |
| Severe stunting | 61 | 34.1 |
| Missing data | 5 | 2.8 |
| **Wasting status at closest visit to 11 months of age** | | |
| Not wasted (Normal WLZ) | 134 | 74.9 |
| Moderate wasting | 32 | 17.9 |
| Severe wasting | 9 | 5.0 |
| Missing data | 4 | 2.2 |
| **Underweight status at closest visit to 11 months of age** | | |
| No underweight (Normal WAZ) | 55 | 30.7 |
| Moderate underweight | 58 | 32.4 |
| Severe underweight | 66 | 36.9 |
| **Stunting status at closest visit to 11 months of age** | | |
| Moderate stunting | 79 | 44.1 |
| Severe stunting | 100 | 55.9 |
| **History of feeding difficulties** | | |
| No | 103 | 57.5 |
| Yes | 21 | 11.7 |
| Missing data | 55.0 | 30.7 |
| **Total number of PDC visits between 11 and 24 months of age (median, IQR)** | 5 | 4, 7 |

[a]Other conditions include the following: central nervous system infections, trisomy 21, post-hospitalization for severe malnutrition when < 12 months of age, hydrocephalus, cleft lip or palate and other developmental delays.
[b] multiple conditions include: being diagnosed with more than one condition
Abbreviations: HIE, hypoxic ischemic encephalopathy; IQR, interquartile range; LBW, low birth weight; PDC, Pediatric Development Clinic; LAZ, length-for-age z-scores; WLZ, weight-for-length z-scores; WAZ, weight-for-age z-scores.

In the bivariate analysis, normal LAZ at 6 months of age, and normal WAZ at 6 and 11 months of age positively associated with recovery from stunting, while, having no health insurance and increasing of maternal age were negatively associated with recovery (S2 Table). Lack of health insurance was completely predictive of failure to recover; no children without health insurance recovered from stunting.

**Table 3. Stunting dynamics at 24 months of children enrolled in PDC in Eastern Province, Rwanda between 2014 and 2018.** N = 179.

| Stunting status at 11 months | Stunting status at 24 months | | |
|---|---|---|---|
| | **Not stunted** | **Moderate** | **Severe** |
| Moderate (n = 79; 44.1%) | 29 (36.7%) | 30 (38.0%) | 20 (25.3%) |
| Severe (n = 100; 55.9%) | 8 (8.0%) | 21 (21.0%) | 71 (71.0%) |
| **Total (N = 179)** | **37 (20.7%)** | **51 (28.5%)** | **91 (50.8%)** |

**Table 4. Factors associated with stunting recovery at 24 months of age for PDC children enrolled in PDC children who were stunted at 11 months of age.**

| Factors | Stunting recovery at 24 months of age | | | | | |
| --- | --- | --- | --- | --- | --- | --- |
| | Univariate analysis | | Full model[b] | | Reduced model[b] | |
| | OR (95% CI) | p-value | aOR (95% CI) | p-value | aOR (95% CI) | p-value |
| **Child's sex** | | 0.340 | | 0.683 | | 0.628 |
| Male | ref | | ref | | ref | |
| Female | 1.4 (0.7–3.0) | | 1.2 (0.5–3.0) | | 1.2 (0.5–2.8) | |
| **Age of the primary caregiver (in years) at the child's enrollment in PDC** | 1.0 (1.0–1.1) | 0.116 | 1.1 (1.0–1.2) | 0.036 | | |
| **Stunting status at closest visit to 6 months of age** | | <0.001 | | 0.302 | | 0.035 |
| Not stunted (Normal LAZ) | ref | | ref | | ref | |
| Moderate Stunting | 0.3 (0.1–0.7) | | 0.5 (0.2–1.4) | | 0.4 (0.16–0.97) | |
| Severe Stunting | 0.1 (0.04–0.4) | | 0.4 (0.08–1.9) | | 0.2 (0.07–0.81) | |
| **Stunting status at closest visit to 11 months of age** | | <0.001 | | 0.006 | | 0.004 |
| Moderate Stunting | ref | | ref | | ref | |
| Severe Stunting | 0.1 (0.1–0.4) | | 0.2 (0.07–0.63) | | 0.3 (0.1–0.6) | |
| **The child was diagnosed with other conditions[a]** | | 0.152 | | 0.168 | | |
| No | ref | | ref | | | |
| Yes | 3.1 (0.7–14.5) | | 3.4 (0.6–19.6) | | | |
| **Underweight status at closest visit to 6 months of age** | | 0.002 | | 0.380 | | |
| No underweight (Normal WAZ) | ref | | ref | | | |
| Moderate Underweight | 0.6 (0.3–1.3) | | 0.9 (0.3–3.0) | | | |
| Severe Underweight | 0.1 (0.04–0.4) | | 0.3 (0.05–1.8) | | | |
| **Underweight status at closest visit to 11 months of age** | | 0.015 | | 0.343 | | |
| No underweight (Normal WAZ) | ref | | ref | | | |
| Moderate Underweight | 0.5 (0.2–1.3) | | 0.5 (0.15–1.8) | | | |
| Severe Underweight | 0.2 (0.1–0.6) | | 1.4 (0.3–5.8) | | | |

[a]Other conditions include the following: central nervous system infections, trisomy 21, post-hospitalization for severe malnutrition when < 12 months of age, hydrocephalus, cleft lip or palate and other developmental delays.

[b]Multivariable logistic regression model using backward stepwise procedures to reduce from full model

Abbreviations: OR, odds ratio; aOR, adjusted odds ratio; CI, confidence interval; LAZ, length-for-age z-scores; WAZ, weight-for-age z-scores

In the final logistic regression model, stunting recovery at 24 months was significantly associated with stunting status at 6 months of age and severity of stunting at age 11 months when controlling for the sex of the child (Table 4).

Early stunting at 6 months of age was associated with lower odds of stunting recovery, with the odds of stunting recovery being reduced by 80% (aOR: 0.2; 95%CI: 0.07–0.81) for severely stunted children and by 60% (aOR: 0.4; 95% CI: 0.16–0.97) for moderately stunted children compared to children with normal height-for-age at 6 months of age (p = 0.035). Lower odds of stunting recovery were observed among children who were severely stunted at 11 months of age (aOR: 0.3; 95%CI: 0.1–0.6) compared to these who were moderately stunted (p = 0.004).

## Discussion

Nearly a quarter (21%) of children in this study who were stunted at 11 months recovered from stunting by age 24 months and factors associated with failure to recover from stunting included early stunting at 6 months of age and greater severity of stunting at age 11 months. This is perhaps the first study in Rwanda evaluating stunting recovery among children born small and sick.

The rates of recovery from stunting in our study are considerable given that the observed changes took place over the course of 12 months. While more than half of children were severely stunted at 11 months of age, recovery occurred in the critical 1,000 days' window when the brain is most rapidly developing. The existing literature on stunting recovery reports variable rates of recovery from stunting depending on the age ranges of children at the time stunting recovery is assessed [18, 28]. A longitudinal study of children from four low- and middle-income countries (Ethiopia, India, Peru, and Vietnam) found stunting recovery rates of 27% to 53% and 30% to 47% at 5 and 8 years old, respectively [28]. Findings from other studies reported stunting recovery rates of 24% in Brazil [18], 35% in India [29] and 19% in South Africa [19]. Our results should be compared with caution to rates from these other studies, however. Most evaluated catch-up growth status of older children outside the 1,000 days' window of critical brain development, had relatively longer timeframes for recovery, and were not conducted among similarly high-risk children [4, 30], Other studies included a general population of children [18, 19, 28, 29] and had no specific interventions to improve catch-up.

While interventions for catch-up growth are important, recovery from stunting may not be sufficient to undo the early negative impact on the brain occurring when children experience chronic undernutrition and poor growth in early life [31, 32]. Improving maternal nutrition and early integrated nutritional and caregiving interventions, including all components of nurturing care for children born small and sick, are crucial for accelerating growth catch-up and improving developmental outcomes [33]. The stunting recovery rates in our study point to the positive impact of the PDC's integrated interventions on nutritional status, and provide important learning on potential strategies that can work to achieve the global nutrition targets for reduction of stunting by 40% by 2025 [34]. Further study could help to understand potential associated benefits for children's development.

In our study, children who were severely stunted at 11 months were much less likely to recover completely at 24 months than their peers with moderate stunting. This finding is consistent with previous studies which reported lower likelihood of recovery from stunting among children with severe stunting [29, 35]. However, in our study, at least 29% of severely stunted children improved from severe to moderate stunting in 12 months while in the PDC program and longer evaluation could help to understand their longer-term growth.

Similarly to findings from the South African study [19], children with early onset of stunting in the first 6 months of life were significantly less likely to recover from stunting. The first 6 months of life is a critical period with fast growth velocity [36]; children exhibiting early linear growth deficiencies may experience slow growth catch-up taking longer for them to recover and may develop persistent stunting in late childhood. The PDC program includes specific early interventions focused on prevention of stunting in the first 6 months of life. In 2017, these interventions were strengthened to improve the quality of nutritional assessment, the PDC clinical protocol and tools were revised with full integration of a tool for management of small and nutritionally at-risk infants under six months and their mothers (MAMI care pathway package) [37]. In addition, decision-making support algorithms were added to provide guidance to PDC providers on appropriate systematic screening, early identification, and management of underweight, wasting and feeding issues for small and nutritionally at risk infants and their mothers [20]. Based on the timing of our study, not all children followed benefited from these revised procedures and future studies could investigate whether these early interventions reduce stunting at 6 months and contribute to overall greater stunting recovery in the PDC.

In our study, only prior stunting status was statistically significantly associated with recovery in the final model. Other factors such as maternal age, and underweight status were not independently associated with recovery when prior stunting is in the model. In our study, the

maternal age in those who recovered was older, but this difference was not significant, while in a study from Kenya found that children from younger mothers were less likely to recover from stunting [17]. This discrepancy may, in part, exist because the age distribution in our study was different than the Kenyan study, which had both larger study population and a higher proportion of adolescent mothers. Insurance status has been shown in other studies to be an important predictor of care seeking [38–40] and associated with lower odds of stunting [41], and in our study lack of insurance completely predicted the outcome, resulting in complete separation of the model, so was not included in the final model. This finding is slightly surprising, as PDC care is free of charge, mothers in lower Ubudehe categories are given travel subsidies as part of social support provided to them and a high proportion (74%) of the uninsured were in low Ubudehe categories. This suggests that PDC services, while important and targeted to remove financial barriers to participation, are not entirely a replacement for other needed health care that may be lacking.

Despite the high percentage of children whose stunting status improved, a quarter of moderately-stunted children worsened to severe stunting, despite PDC interventions. Similar probabilities of progressing from moderate to severe stunting have been reported from a longitudinal study conducted in Ethiopia [42]. Nevertheless, our finding demonstrates how it can be difficult to reverse stunting given the multi-sectoral factors that contribute to stunting [43, 44] and challenges to achieve full stunting recovery among infants and young children enrolled in PDC in just a 12-month period. Our findings also imply a need for further qualitative research to understand problems faced by caregivers and PDC providers in supporting children with poor growth and solutions to strengthen the program interventions as well as other types of support to caregivers. It is also possible that observed worsening and challenges to achieve stunting recovery among children in this study could be related to other co-morbidities and challenges. For example, feeding difficulties are common among children with developmental disabilities and tend to worsen without early intervention. Feeding difficulties among children with disabilities are associated with high rates of malnutrition and early life mortality [45]. Previous studies conducted in Rwanda among children born small and sick reported high rates of potential disabilities at ages 2 to 4 years [46] and higher rates of malnutrition among children with feeding difficulties [47], thus additional specialized nutrition interventions are likely required to address these children's needs. Based on this challenge, in 2019, the PDC program integrated early intervention using the Baby Ubuntu model [48], a group participatory early intervention program starting from 6 months for young children with disability and their families. The 10-module, participatory, parents' peer support group program includes a module designed to provide caregivers with skills to appropriately feed a child with a disability and manage their feeding difficulties at home [49, 50].

Our study had certain limitations to note. First, as our study used data that were routinely collected from the EMR there were some variables of interest that were not available, such as information regarding the home environment including water, sanitation and hygiene; food security; child care practices at home; and female decision-making in households. However, some important socio-demographic variables were available in the EMR which we included, such as household socioeconomic status (Ubudehe category). Second, as these were routinely collected programmatic data, missing data was an issue for many different variables. Our sensitivity analyses for systematic differences between children who were eligible at 11 months and included versus those who were excluded because of lack of data at 24 months found no statistically significant differences in any sociodemographic or baseline health factors, so we do not believe that this exclusion introduced important biases. For our outcomes measures, we used z scores with cutoff points as determined by WHO and international

studies; using a continuous z score would have allowed for more nuance in assessment, as the clinical difference between (for example) -2.01 and -1.99 is minimal. However, the utilization of cutoffs is the basis for provision of certain services and additionally allows us to compare to other surveys and population. Also, our study participants were children enrolled in PDC. thus, generalizability of findings to other high-risk infants who do not receive similar interventions is limited. Odds ratios may not be identical to risk in a population with common outcomes, so in our study we present odds exclusively. Finally, our study looked at change over only 12 months, which may have been too short of a timespan to detect important changes. However, since this is among the first studies conducted on a programmatic intervention targeting vulnerable children in a rural African setting, the results remain useful for designing interventions for this special population. Despite these limitations, this study is contributing to the understanding of the effects of an integrated program and early interventions on growth-catch-up among high-risk infants, and the extent of stunting recovery possible with supportive programming among infants born with perinatal complications at risk of malnutrition and developmental challenges.

## Conclusion

Our study findings generate new evidence about the recovery from stunting among at infants and young children enrolled in the PDC in rural Rwandan settings. A substantial proportion of children enrolled in the PDC that were stunted at 11 months of age recovered from stunting at 24 months of age despite their history of perinatal complications. Early stunting at 6 months of age and severity of stunting at 11 months of age were negatively associated with stunting recovery at 24 months of age. Our results highlight the importance of integrated interventions to prevent undernutrition during first two years of life and observed stunting recovery rates provide promising effects of integrated and targeted nurturing care interventions to avert stunting during infancy and early childhood periods among at risk infants with history of perinatal complications. Further research is needed to explore association of environmental factors with stunting recovery at 24 months of age and in late childhood among high-risk children enrolled in post-neonatal follow-up programs. The effects of sociodemographic factors of both parents including parental stature need also to be included in future studies. In addition, other studies including controls groups could help to understand the effectiveness of PDC interventions on catch-up growth and provide more evidence to guide improvements as well as influencing policies.

## Supporting information

**S1 Checklist. Inclusivity in global research.**
(DOCX)

**S1 Table. Comparing the characteristics of PDC children who were stunted at age 11 months by their eligibility status for inclusion in the analysis for stunting recovery at age 24 months, N = 403 unless otherwise indicated.**
(DOC)

**S2 Table. Bivariate analysis of the factors associated with stunting recovery at 24 months of age among PDC children identified with stunting at age of 11 months, N = 179 unless otherwise indicated.**
(DOC)

## Acknowledgments

We acknowledge Partners In Health for the support of this work. This study was developed under the Partners In Health/Inshuti Mu Buzima Intermediate Operational Research Training Program, developed and facilitated by Dale A. Barnhart, Bethany Hedt-Gauthier and Ann C. Miller. Ann C. Miller and Alphonse Nshimyiryo provided direct mentorship to this paper as part of this training. We thank Kirehe and Rwinkwavu district hospitals and their health centers for their support during this study. We also acknowledge the contributions of PDC nurses, social workers and EMR data officers for data collection and for their daily efforts to improve the lives of infants and young children in rural communities. We are grateful to the PDC children and their caregivers in the study.

## Author Contributions

**Conceptualization:** Mathieu Nemerimana, Silas Havugarurema, Alphonse Nshimyiryo, Catherine M. Kirk, Kathryn Beck, Ann C. Miller.

**Data curation:** Mathieu Nemerimana, Alphonse Nshimyiryo, Ann C. Miller.

**Formal analysis:** Mathieu Nemerimana, Alphonse Nshimyiryo, Ann C. Miller.

**Funding acquisition:** Catherine M. Kirk.

**Investigation:** Mathieu Nemerimana.

**Methodology:** Mathieu Nemerimana, Alphonse Nshimyiryo, Angelique Charlie Karambizi, Catherine M. Kirk, Kathryn Beck, Ann C. Miller.

**Project administration:** Mathieu Nemerimana.

**Software:** Todd Anderson.

**Supervision:** Alphonse Nshimyiryo, Catherine M. Kirk, Kathryn Beck, Ann C. Miller.

**Validation:** Mathieu Nemerimana, Silas Havugarurema, Alphonse Nshimyiryo, Angelique Charlie Karambizi, Chantal Gégout, Olivier Bigirumwami, Ann C. Miller.

**Visualization:** Mathieu Nemerimana.

**Writing – original draft:** Mathieu Nemerimana.

**Writing – review & editing:** Mathieu Nemerimana, Silas Havugarurema, Alphonse Nshimyiryo, Angelique Charlie Karambizi, Catherine M. Kirk, Kathryn Beck, Chantal Gégout, Todd Anderson, Olivier Bigirumwami, Jules Maurice Ubarijoro, Patient K. Ngamije, Ann C. Miller.

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
