## [Decision Letter · Decision Letter 0]

29 Dec 2022

PONE-D-22-00485Factors associated with recovery from stunting at 24 months of age among infants and young children enrolled in the Pediatric Development Clinic (PDC): a retrospective cohort study in rural RwandaPLOS ONE

Dear Dr. Nemerimana, 

Thank you for submitting your manuscript to PLOS ONE. After careful consideration, we feel that it has merit but does not fully meet PLOS ONE’s publication criteria as it currently stands. Therefore, we invite you to submit a revised version of the manuscript that addresses the points raised during the review process.

Please address the comments from the two reviewers. Please also get your manuscript thoroughly edited for English.Please ensure that your decision is justified on PLOS ONE’s publication criteria and not, for example, on novelty or perceived impact.

We look forward to receiving your revised manuscript.

Kind regards,

Gouranga Lal Dasvarma, PhD

Academic Editor

PLOS ONE

Journal Requirements:

2. Please include a complete copy of PLOS’ questionnaire on inclusivity in global research in your revised manuscript. Our policy for research in this area aims to improve transparency in the reporting of research performed outside of researchers’ own country or community. The policy applies to researchers who have travelled to a different country to conduct research, research with Indigenous populations or their lands, and research on cultural artefacts. The questionnaire can also be requested at the journal’s discretion for any other submissions, even if these conditions are not met.  Please find more information on the policy and a link to download a blank copy of the questionnaire here: https://journals.plos.org/plosone/s/best-practices-in-research-reporting. Please upload a completed version of your questionnaire as Supporting Information when you resubmit your manuscript.”

3. Please ensure that you have specified (1) whether consent was informed, (2) what type you obtained (for instance, written or verbal, and if verbal, how it was documented and witnessed). If your study included minors, state whether you obtained consent from parents or guardians. If the need for consent was waived by the ethics committee and (3) If you are reporting a retrospective study of medical records or archived samples, please ensure that you have discussed whether all data were fully anonymized before you accessed them and/or whether the IRB or ethics committee waived the requirement for informed consent. If patients provided informed written consent to have data from their medical records used in research, please include this information.

“During the study period, the PDC program has been supported with funding through Grand Challenges Canada Saving Brains and the Primates World Relief and Development Fund via Global Affairs Canada.

“During the study period, the PDC program has been supported with funding through Grand Challenges Canada Saving Brains and the Primates World Relief and Development Fund via Global Affairs Canada. We thank Kirehe and Rwinkwavu district hospitals and their health centers for supporting PDC implementation. We also acknowledge the contributions of PDC nurses, social workers and EMR data officers for data collection and for their daily efforts to improve the lives of infants and young children in rural communities. We are grateful to the PDC children and their caregivers in the study.  “

“During the study period, the PDC program has been supported with funding through Grand Challenges Canada Saving Brains and the Primates World Relief and Development Fund via Global Affairs Canada.

Additional Editor Comments (if provided):

In view of the continuing delay in getting a second reviewer of this manuscript, I have invited myself as the second reviewer as permissible under PLOS One policy and completed the review recommending a major revision.

Reviewers' comments:

Reviewer's Responses to Questions

**Comments to the Author**

1. Is the manuscript technically sound, and do the data support the conclusions?

Reviewer #1: Partly

Reviewer #2: Partly

2. Has the statistical analysis been performed appropriately and rigorously? 

Reviewer #1: No

Reviewer #2: No

3. Have the authors made all data underlying the findings in their manuscript fully available?

Reviewer #1: Yes

Reviewer #2: Yes

4. Is the manuscript presented in an intelligible fashion and written in standard English?

Reviewer #1: No

Reviewer #2: No

5. Review Comments to the Author

Reviewer #1: The paper's stated goal is to assess the proportion of children who recover from stunting at 24 months and the factors associated with that recovery. While the premise is reasonable, and the authors have done a great job analyzing and writing this up, some changes are needed:

- Logistic regression should be used when outcomes are rare. In this case, the outcome is not rare (~20% recovered). Some related articles: https://jamanetwork.com/journals/jama/fullarticle/188182

https://stats.oarc.ucla.edu/stata/faq/how-can-i-estimate-relative-risk-using-glm-for-common-outcomes-in-cohort-studies/

- The paper could be tightened up considerably. It reads like a thesis, but should stay focused on the primary question of interest. (are there factors that are associated with recovery from stunting between 11 and 24m?) The tables are too heavy (things that are in the tables should be directly relevant to those factors included in Table 5 - remove all of the other variables that get dropped from the analysis, and if you really want to show that work, include as supplementary material).

- It seems like in Table 4 you are using chi squared (or something?) to get those p-values? If that table is to be retained, it should be put in supplementary material. (also in Table 4 - the categories for 'Stunting status at closest visit to 11 months' are 'Moderate underweight' and 'Severe underweight' - should be stunting, not underweight.

- Table 5 - I would put the unadjusted effects of each of the variables in the first columns, then the full model, then the reduced model. Is the 'full model' a multivariable model that includes all of the variables listed there? It's a bit unclear.

- In general, tables and figures should be 'stand-alone' and should provide enough information to the reader to be able to understand what they are looking at without reading the paper. Please provide more details in the table and figure legends. (eg, what sort of model, etc.)

- Seems to me that stunting at 6m and stunting at 11m would be highly correlated, and I'm not sure what we gain from the inclusion of the 6m variable. It seems like the authors could do a separate analysis looking at the 6m stunting variable in order to look at that relationship?

- The authors should highlight the finding of 0 children without health insurance recovering. (underscores that health insurance is important)

- I see that the authors cited Leroy so I am sure they are aware of the disadvantages of using a categorical outcome for a continuous measure. It would be great to see them use the continuous Z-scores in some way, or at least address that moving from -2.01 to -1.99, while technically moving from stunted to not stunted, is not a huge improvement.

- The introduction and setting sections were a bit repetitive, and the setting paragraph went into too much detail about the services provided at the centers. The discussion was also too long.

- In the results, it felt a little too much like the authors just wrote in the text what was in the tables. It is best to highlight some interesting findings and let the reader look at the table for the other details.

- On line 235, 132 (72.7%) were stunted (not 73.7%)

- These are children who were sick/LGA very early and have been going to these centers about once every 2-3 months. Did you look at the relationship between number of visits and recovery from stunting? Do you think the kids who went more often were those who were doing worse, or did they actually do better because of more contact with the health center? Do you have any comparable outcomes of kids with similar conditions in areas without these centers and what percent of those kids are stunted at 2 years?

Reviewer #2: Reviewer’s comments.

PONE-D-22-00485: “Factors associated with stunting recovery… “

by Mathew Nemerimana et al

Reviewer: Gouranga Dasvarma

General comments:

This article traces a small sample of children who were stunted at 11 months of age and recovered from stunting at 24 months of age and identifies the factors associated with such recovery. The authors have used stringent criteria to select the cases for inclusion in the study, that saw an initially recruited 1,875 children being reduced to a small sample of only 179 “eligible” children for study after excluding 1,678 (90.4%) children from the initial sample. Under the circumstances one might wonder how generalisable the findings would be for the population at large. Although this limitation is acknowledged by the authors (lines 372-379 of the manuscript), the stated justification for the study (lines 380-386) does not sound strong enough about its replicability in the general population.

Re data analysis, Table 4 presumably shows the results of Chi-square analyses, which should be clearly mentioned. Further, the Chi-square values (as well as the p-values) need to be shown in order to know the strength of association between the predictor variables and stunting recovery at 24 months. The p-values show only whether the observed associations are statistically significant but not how strong they are. At least, the Chi-square values for the statistically significant associations should be shown.

Table 5: The rationale for using logistic regression should be discussed beforehand. Moreover, even if the use of logistic regression is justified, the predictor variables, “Stunting status at closest to 6 months of age” and “Stunting status at closest to 11 months of age” may be correlated with each other. Has a collinearity test been done? A similar question may be asked about the predictor variables: underweight status at 6 and 11 months of age.

The Discussion section is too wordy. Please reduce its length by keeping the essential points. Consider removing repetitions of results and presenting the findings of other studies in a succinct manner.

Specific comments:

Abstract.

1. Results: What were the socio-economic factors associated with stunting recovery (refer to the last sentence of the Methods section of the abstract)

2. Lines 35-36. State briefly in the Results section what influence did the socio-demographic and clinical characteristics of the child’s primary caregiver have on stunting recovery?

3. Lines 46-49. Did the 179 children stay enrolled in the PDC for 24 months or more?

4. Lines 92-94. Please re-write the aim more clearly. Here is a suggestion:

“The study aimed to examine, among children recruited in rural PDCs of Rwanda and stunted at 11 months what proportion recovered at 24 months of age and which factors were associated with their recovery”.

5. Line 95. Methods. Is there any information (data) about the number of children who are stunted at 11 months and how many of them recovered at 24 months?

6. Setting and Intervention (Lines 100-130) and Study Population (Lines 131-143)

These two sub-sections warrant separate sections by themselves, and not as a part of

7. METHODS. State the rationale for choosing 10 PDCs.

8. METHODS (Lines 95-99). Move this to just before Data Collection.

9. Line 148. Who were the primary caregivers?

10. Line 171. ., “we dichotomised…”. Is dichotomised the right word? Can you not say “categorised”?

11. Lines 188-193. What is the proportion of pre-term births?

12. Lines 209-210. Are those factors chosen in the final model shown in the table?

13. RESULTS (Lines 220-238). It would be desirable to interpret the findings of Table 1 instead of describing them, which everyone can see. Moreover, there are limited data on the socio-demographic characteristics of children and their caregivers (presumably caregivers are the children’s mothers) – data on children’s household sanitation and data on birth interval would have been very useful.

14. Lines 244-253. It is difficult to identify the results in Table 4 from their description given in the text. For example, “Stunting recovery … with normal LAZ at 6 moths” (lines 244-243) corresponds to line 62-64 of Table 4, therefore it would be helpful to add within brackets “(normal LAZ)” after “Not Stunted”.

15. Line 306. The meaning of the sentence is not clear.

16. Lines 399-401. Re further research, Socio-demographic factors of parents, parental education and parental stature should be included in future studies.

17. Please discuss how your findings compare with the WHO’s target of reducing stunting among children (https://www.who.int/publications/i/item/WHO-NMH-NHD-14.3).

18. Table 4. Lines74-76. “Stunting status at closest visit to11 months”. The sub-categories shown here are: “Moderate Underweight” and “Severe Underweight”. Should these not refer to stunting status, rather than underweight status?

19. Lines 393-399. “Our results emphasize the need ….development potential”. These needs are not directly evident from the results of the present study, rather they appear to reflect the authors’ prior knowledge of these matters. Therefore, these sentences should be appropriately modified, with necessary references if needed.

6. PLOS authors have the option to publish the peer review history of their article (what does this mean?). If published, this will include your full peer review and any attached files.

Reviewer #1: No

Reviewer #2: No

---

## [Author Response · Author response to Decision Letter 0]

7 Mar 2023

We thank the reviewers for their insightful comments and for the opportunity to revise and strengthen this manuscript. We have responded to all comments and made revisions of the manuscript to improve the quality of the manuscript. Our changes in the revised manuscript are highlighted with track changes. We hope the revisions in the manuscript and our responses to comments will satisfy the reviewers and editor.

Find below our responses to each point raised by editor and reviewers’ comments.

Journal requirements

We have reviewed our manuscript and made changes where required to meet PLOS ONE’S requirements.

2. Please include a complete copy of PLOS’ questionnaire on inclusivity in global research in your revised manuscript. Our policy for research in this area aims to improve transparency in the reporting of research performed outside of researchers’ own country or community. The policy applies to researchers who have travelled to a different country to conduct research, research with Indigenous populations or their lands, and research on cultural artefacts. The questionnaire can also be requested at the journal’s discretion for any other submissions, even if these conditions are not met. Please find more information on the policy and a link to download a blank copy of the questionnaire here: https://journals.plos.org/plosone/s/best-practices-in-research-reporting. Please upload a completed version of your questionnaire as Supporting Information when you resubmit your manuscript.” 

The checklist has been completed and a “Inclusivity in global research” section has been added to the methods on page 11 

3. Please ensure that you have specified (1) whether consent was informed, (2) what type you obtained (for instance, written or verbal, and if verbal, how it was documented and witnessed). If your study included minors, state whether you obtained consent from parents or guardians. If the need for consent was waived by the ethics committee and (3) If you are reporting a retrospective study of medical records or archived samples, please ensure that you have discussed whether all data were fully anonymized before you accessed them and/or whether the IRB or ethics committee waived the requirement for informed consent. If patients provided informed written consent to have data from their medical records used in research, please include this information. 

Our study was a retrospective study of electronic medical records and the data were de-identified prior conducting analysis. The requirement for written informed consent was waived this is now specified in the section of ethical considerations on page 11

4. Thank you for stating in your Funding Statement: “During the study period, the PDC program has been supported with funding through Grand Challenges Canada Saving Brains and the Primates World Relief and Development Fund via Global Affairs Canada. The funders had no role in study design, data collection and analysis, decision to publish, or preparation of the manuscript.” Please provide an amended statement that declares *all* the funding or sources of support (whether external or internal to your organization) received during this study, as detailed online in our guide for authors at http://journals.plos.org/plosone/s/submit-now. Please also include the statement “There was no additional external funding received for this study.” in your updated Funding Statement. Please include your amended Funding Statement within your cover letter. We will change the online submission form on your behalf.

We have revised our statement on funding. Below is the updated statement:

“During the study period, the PDC program has been supported with funding through Grand Challenges Canada Saving Brains (R-SB-POC-1707-09583), the Primates World Relief and Development Fund via Global Affairs Canada (D-001975-001) and Partners In Health/Inshuti Mu Buzima. There was no additional external funding received for this study. The funders had no role in study design, data collection and analysis, decision to publish, or preparation of the manuscript.”

5. We note that the grant information you provided in the ‘Funding Information’ and ‘Financial Disclosure’ sections do not match. When you resubmit, please ensure that you provide the correct grant numbers for the awards you received for your study in the ‘Funding Information’ section.

Grant number was added on the funding information. 

6. Thank you for stating the following in the Acknowledgments Section of your manuscript: “During the study period, the PDC program has been supported with funding through Grand Challenges Canada Saving Brains and the Primates World Relief and Development Fund via Global Affairs Canada. We thank Kirehe and Rwinkwavu district hospitals and their health centers for supporting PDC implementation. We also acknowledge the contributions of PDC nurses, social workers and EMR data officers for data collection and for their daily efforts to improve the lives of infants and young children in rural communities. We are grateful to the PDC children and their caregivers in the study.” We note that you have provided additional information within the Acknowledgements Section that is not currently declared in your Funding Statement. Please note that funding information should not appear in the Acknowledgments section or other areas of your manuscript. We will only publish funding information present in the Funding Statement section of the online submission form. Please remove any funding-related text from the manuscript and let us know how you would like to update your Funding Statement. Currently, your Funding Statement reads as follows: “During the study period, the PDC program has been supported with funding through Grand Challenges Canada Saving Brains and the Primates World Relief and Development Fund via Global Affairs Canada. The funders had no role in study design, data collection and analysis, decision to publish, or preparation of the manuscript.” Please include your amended statements within your cover letter; we will change the online submission form on your behalf.

We have revised our statement on funding information. The updated statement is provided above and it is included in the cover letter. As you recommended, we have removed all funding related information from the acknowledgements section and corrected the funding statement as noted above.

Reviewers’ comments:

REVIEWER#1

1. The paper's stated goal is to assess the proportion of children who recover from stunting at 24 months and the factors associated with that recovery. While the premise is reasonable, and the authors have done a great job analyzing and writing this up, some changes are needed 

Thank you for the review and important comments to improve the quality of the manuscript 

2. Logistic regression should be used when outcomes are rare. In this case, the outcome is not rare (~20% recovered). Some related articles: https://jamanetwork.com/journals/jama/fullarticle/188182
https://stats.oarc.ucla.edu/stata/faq/how-can-i-estimate-relative-risk-using-glm-for-common-outcomes-in-cohort-studies/

We thank the reviewer for this thought-provoking suggestion. We agree that logistic regression produces odds ratios that are a good approximation of risk ratio when outcomes are rare, but respectfully disagree that they must be discarded when the outcome is not. An odds ratio is itself a valid measure, even when it does not closely approximate the risk ratio. We are careful to express in the text that we present relative odds; not risk. Additionally, some of the papers we cite as comparisons use similar measures with similarly non-rare endpoints (see Lu et al.). We did run these models in Stata, and have not seen differences in factors statistically significantly associated with the outcome, although we have seen differences in strength of association. We have added a sentence in the limitations section to address this reviewer’s concern as follows: 

Odds ratios may not be identical to risk in a population with common outcomes, so in our study we present odds exclusively.

3. The paper could be tightened up considerably. It reads like a thesis, but should stay focused on the primary question of interest. (are there factors that are associated with recovery from stunting between 11 and 24m?) 

Thank you for the recommendation. We have revised the main text of manuscript to tighten it up. 

4. The tables are too heavy (things that are in the tables should be directly relevant to those factors included in Table 5 - remove all of the other variables that get dropped from the analysis, and if you really want to show that work, include as supplementary material).

We have reduced the number of tables and put Table 4 which was presenting results for bivariate analysis in the supplementary materials 

5. It seems like in Table 4 you are using chi squared (or something?) to get those p-values? If that table is to be retained, it should be put in supplementary material.

We have used Fisher’s exact test for categorical variables and Wilcoxon Rank Sum test for continuous variables in bivariate analysis to get the p-values in Table 4. Footnotes describing the tests used have been added on Table 4. Fisher’s exact test was chosen to be used over Chi-square test since some categories in some variables had small counts. As suggested, we have put the original Table 4 as supplementary material (S2 Table)

6. (also in Table 4 - the categories for 'Stunting status at closest visit to 11 months' are 'Moderate underweight' and 'Severe underweight' - should be stunting, not underweight.

The categories for stunting status at closest visit to 11 months are correctly changed to 'Moderate Stunting' and 'Severe Stunting'

7. Table 5 - I would put the unadjusted effects of each of the variables in the first columns, then the full model, then the reduced model. Is the 'full model' a multivariable model that includes all of the variables listed there? It's a bit unclear. 

Thank you for the recommendation. We have added a column on the tables that show unadjusted effects of each variable in Table 5 (which is now table 4). 

The full model was multivariable logistic regression model, we have added footnote on the table to specify this.

8. In general, tables and figures should be 'stand-alone' and should provide enough information to the reader to be able to understand what they are looking at without reading the paper. Please provide more details in the table and figure legends. (eg, what sort of model, etc.) 

Thank you for the recommendations. We have added additional details on the tables and figure, such as a legend for abbreviations as well as details on the analysis methods. 

9. Seems to me that stunting at 6m and stunting at 11m would be highly correlated, and I'm not sure what we gain from the inclusion of the 6m variable. It seems like the authors could do a separate analysis looking at the 6m stunting variable in order to look at that relationship?

We do agree with the reviewer that stunting at 6 month and stunting at 11 months of age are correlated. However since infants who get early stunting in their first 6 months of age do not have the same growth catch-up trajectories as those who get stunting at a later age, it was important to investigate the effects of being stunted at 6 months of age on the stunting recovery. In our analysis stunting at 6 months of age and the severity of stunting at 11 months were among the predictors of stunting recovery.

10. The authors should highlight the finding of 0 children without health insurance recovering. (underscores that health insurance is important) Thank you for the recommendation. We have highlighted with underscores the finding 0 of children without health insurance recovering from stunting. We have included the following sentence in our results section: “Lack of health insurance was completely predictive of failure to recover; no children without health insurance recovered from stunting.”

In our study we defined recovery from stunting as moving from length-for-age Z-scores (LAZ) <-2 to >=2, this was based on the 2006 WHO growth standards cut points on definition of stunting. In our context, these cut points are the basis for access to services and are used globally to estimate status of children affected by stunting. Hence, in this study we based on the WHO growth standards cut points. We agree with the reviewer that moving from -2.01 to -1.99 is not huge improvement but it demonstrates a positive change in the status of a child’s linear growth, particularly in children with high risk of developmental challenges this change while small can require a lot of interventions. We do also agree that using continuous Z-scores as an outcome measure would be preferable and can demonstrate different improvement changes that some children would have undergone, for example a child moving from -4 to -3, though they may still be stunted. We used categorical outcome since our study was mainly focused on assessing the proportion of children who exhibited full recovery from stunting. We have added in our discussions section a limitation that using continuous LAZ-scores would capture more detail.

11. The introduction and setting sections were a bit repetitive, and the setting paragraph went into too much detail about the services provided at the centers. The discussion was also too long.

We have revised both introduction and setting sections to remove repetitive statements. The discussions was also shortened.

12. In the results, it felt a little too much like the authors just wrote in the text what was in the tables. It is best to highlight some interesting findings and let the reader look at the table for the other details.

Thank you for the observation we have revised the section of results to focus on only key findings.

13. On line 235, 132 (72.7%) were stunted (not 73.7%)

Thank you for the observation. We have reviewed and corrected table 2, percentages on moderate stunting 71 (39.7%). We maintained description of 132 (73.7%), which is equivalent to: 132 out of 179 children were stunted at closest visit to 6 months of age.

14. These are children who were sick/LGA very early and have been going to these centers about once every 2-3 months. Did you look at the relationship between number of visits and recovery from stunting? Do you think the kids who went more often were those who were doing worse, or did they actually do better because of more contact with the health center? 

Yes, in our analysis, we looked at the relationship between total number of visits and recovery from stunting. We found that the median number of visits was higher among children who did not recover (median= 15 visits) compared to those who did recover (median= 12 visits, p<0.001). This finding can be confusing to interpret as the number of visits is determined both by the child’s reason for enrollment in the PDC as well as their current health, nutrition, and developmental needs. Children who have complex conditions such as developmental disabilities and congenital malformations require frequent contact visits to attend different medical services. In addition, children with growth faltering are followed up more frequently to monitor progress. So while one may expect more visits, as an indicator of potential regularity of engagement with the service, to be associated with better outcomes, this is unlikely to be the case in our context. This variable was dropped from analysis given the complexity of interpretation. 

We have provided the results table as an appendix of this letter below (appendix 1). 

15. Do you have any comparable outcomes of kids with similar conditions in areas without these centers and what percent of those kids are stunted at 2 years?

No we do not have comparable kids with similar conditions in the areas without PDCs. Since this was a retrospective cohort study it was not possible to have comparison group. However, we do have a study that was conducted prior to the existence of PDC among children of slightly different ages in these communities which found 71.8% of children ages 1 to 3 years were stunted (Kirk et al., 2017). Because the age ranges are different in this study, we have opted to not compare these findings in our current paper. 

REVIEWER #2

General comments:

1. This article traces a small sample of children who were stunted at 11 months of age and recovered from stunting at 24 months of age and identifies the factors associated with such recovery. The authors have used stringent criteria to select the cases for inclusion in the study, that saw an initially recruited 1,875 children being reduced to a small sample of only 179 “eligible” children for study after excluding 1,678 (90.4%) children from the initial sample. Under the circumstances one might wonder how generalisable the findings would be for the population at large. Although this limitation is acknowledged by the authors (lines 372-379 of the manuscript), the stated justification for the study (lines 380-386) does not sound strong enough about its replicability in the general population.

We thank the reviewer for this question, and appreciate the opportunity to clarify. 

It is true we have included in the flow chart many children who were not eligible for the study, in the interest of transparency. However, only those with documented stunting at 11 months should be considered eligible. Given this reviewer’s concern, we have conducted a sensitivity analysis of the 403 children who were eligible based on 11 mo stunting status; 179 of whom were included in the analyses. We found no statistically significant differences between those included and excluded because of lack of 24 months visit on sex, socioeconomic status, birthweight, small for gestational age (SGA), prematurity, hypoxic ischemic encephalopathy (HIE), feeding difficulty or any of the malnutrition indicators at 6 months. Thus, we feel that the limited population does not indicate substantial selection bias, and are more comfortable that these findings would be generalizable to a similar population of children.

This study was conducted on a specific, targeted population of children with high, special needs and with families expected to have challenges in caring for small and sick infants. The study population were provided specific interventions through follow up, thus we agree with the reviewer that the findings from this study would not be generalizable to general population of children. However, we feel that the findings would be applicable and generalizable to other populations in low-resource, rural settings with similar populations of at-risk children. In fact, as heightened modern medical care becomes (thankfully) more available in lower resource contexts, many more children with low birth weight or other early life challenges are surviving and need services. While we recognize that a larger sample would have been desirable, our study still provides new learning about an important population of children born small or sick who are at greatest risk of poor nutritional and developmental outcomes. 

2. Re data analysis, Table 4 presumably shows the results of Chi-square analyses, which should be clearly mentioned. Further, the Chi-square values (as well as the p-values) need to be shown in order to know the strength of association between the predictor variables and stunting recovery at 24 months. The p-values show only whether the observed associations are statistically significant but not how strong they are. At least, the Chi-square values for the statistically significant associations should be shown.

We have used Fisher’s exact test for categorical variables and Wilcoxon Rank Sum test for continuous variables in bivariate analysis to get the P-values in Table 4. Footnotes describing the tests used have been added on Table 4. Fisher’s exact test was chosen to be used over Chi-square test since some categories in some variables had small counts. This original Table 4 was also moved to supplementary materials based on feedback from the other reviewer. 

3. Table 5: The rationale for using logistic regression should be discussed beforehand. Moreover, even if the use of logistic regression is justified, the predictor variables, “Stunting status at closest to 6 months of age” and “Stunting status at closest to 11 months of age” may be correlated with each other. Has a collinearity test been done? A similar question may be asked about the predictor variables: underweight status at 6 and 11 months of age.

We thank the reviewer for this suggestion. We have provided a rationale for the use of logistic regression in the response to the other reviewer. As this reviewer has suggested, we conducted a collinearity test using a variance inflation factor assessment on the final model and found that the VIFs for 6 month and 11 month stunting were 2.6 and 2.3. The VIFs were below 5, suggesting that it is not a severe enough correlation to require removal of one of the factors.

4. The Discussion section is too wordy. Please reduce its length by keeping the essential points. Consider removing repetitions of results and presenting the findings of other studies in a succinct manner.

Thank you for the recommendation. We have revised the section of discussion and shorted based on your recommendations.

Specific comments:

5. Abstract.

Results: What were the socio-economic factors associated with stunting recovery (refer to the last sentence of the Methods section of the abstract)

In our analysis, there no socio-economic demographic found with statistical significant association with stunting recovery. We have added on the results section of the abstract a statement describing this finding.

6. Lines 35-36. State briefly in the Results section what influence did the socio-demographic and clinical characteristics of the child’s primary caregiver have on stunting recovery?

No sociodemographic and clinical characteristic of the child primary caregiver (mother) was found to be significantly associated with stunting recovery. We have added in the results section as short statement describing this finding.

7. Lines 46-49. Did the 179 children stay enrolled in the PDC for 24 months or more?

Yes, all 179 children stayed enrolled in PDC for the 24 months and after. Enrolment process and duration follow up period for all children enrolled in PDC are described in the description of the intervention in the methods section.

8. Lines 92-94. Please re-write the aim more clearly. Here is a suggestion:

“The study aimed to examine, among children recruited in rural PDCs of Rwanda and stunted at 11 months what proportion recovered at 24 months of age and which factors were associated with their recovery”. 

Thank for your suggestion. We have rewritten the aim to: “to examine, among children enrolled in PDCs in Rural Rwanda and stunted at 11 months, what proportion recovered at 24 months of age and which factors were associated with their recovery.”

9. Line 95. Methods. Is there any information (data) about the number of children who are stunted at 11 months and how many of them recovered at 24 months?

Information on children who were stunted at 11 months is provided in figure 1. Out of 798 children in PDC who were eligible for study, 403 (50.5%) were stunted of whom 179 had a visit with stunting status within +/- 24 months of age were included in this study. As described in the results, out 179 children who were stunted at 11 months, 37 (20.7%) recovered from stunting at their 24 months of age.

10. Setting and Intervention (Lines 100-130) and Study Population (Lines 131-143). These two sub-sections warrant separate sections by themselves, and not as a part of

Thank you for the recommendation. We have separated description of intervention from setting.

11. METHODS. State the rationale for choosing 10 PDCs.

Thank you for your recommendation the rationale for choosing 10 PDCs was added:

“By the time of data collection in 2020, PDC was being implemented at 10 health facilities in Rwanda and all were included in the study.”

12. METHODS (Lines 95-99). Move this to just before Data Collection.

We have moved the sentence to section of data collection.

13. Line 148. Who were the primary caregivers?

The primary caregivers were mothers. We have replaced the term “primary caregivers” with “mothers” where it applies to be consistent.

14. Line 171. ., “we dichotomised…”. Is dichotomised the right word? Can you not say “categorised”? 

Thank you for your suggestion. We have changed the word “dichotomized” to “categorized…”. 

15. Lines 188-193. What is the proportion of pre-term births?

As presented in Table 2, the proportion of children born preterm (<37 weeks of gestation) was 45.2% (n=81). However, there was 52 children without gestational age documented; therefore, their status on prematurity was missing.

16. Lines 209-210. Are those factors chosen in the final model shown in the table?

Yes the factors that were included in the final model are shown in Table 5 (which currently became table 4)

17. RESULTS (Lines 220-238). It would be desirable to interpret the findings of Table 1 instead of describing them, which everyone can see. Moreover, there are limited data on the socio-demographic characteristics of children and their caregivers (presumably caregivers are the children’s mothers) – data on children’s household sanitation and data on birth interval would have been very useful.

We have revised the section of results on with interpretation findings in Table 1. The term of “caregivers” was changed to “mothers”. 

We agree that information on household sanitation and birth interval would have been useful. Since we used retrospective cohort from electronic medical records that does not capture data on household sanitation or home environment, it was not possible to have these data. This was one of the limitations of this study. We have acknowledged this limitation in the discussion section.

18. Lines 244-253. It is difficult to identify the results in Table 4 from their description given in the text. For example, “Stunting recovery … with normal LAZ at 6 moths” (lines 244-243) corresponds to line 62-64 of Table 4, therefore it would be helpful to add within brackets “(normal LAZ)” after “Not Stunted”.

Thank you for the observation. We have added “normal LAZ” within brackets after not stunted to the tables. We also added descriptions “normal WLZ” within brackets after “not wasted” and “normal WAZ” after “no underweight” in the tables.

19. Line 306. The meaning of the sentence is not clear. 

This sentence was removed.

20. Lines 399-401. Re further research, Socio-demographic factors of parents, parental education and parental stature should be included in future studies.

Thank you for your suggestion. We have added the need to include socio-demographic factors of parents in future studies.

21. Please discuss how your findings compare with the WHO’s target of reducing stunting among children (https://www.who.int/publications/i/item/WHO-NMH-NHD-14.3).

Thank you for this helpful reference. We have added the following sentence to the discussion: “The stunting recovery rates in our study point to the positive impact of the PDC’s integrated interventions on nutritional status, and provide important learning on potential strategies that can work to achieve the global nutrition targets for reduction of stunting by 40% by 2025 [34].” 

22. Table 4. Lines74-76. “Stunting status at closest visit to11 months”. The sub-categories shown here are: “Moderate Underweight” and “Severe Underweight”. Should these not refer to stunting status, rather than underweight status?

We have corrected sub-categories on stunting status at closest visit to 11 months to “moderate stunting” and “severe stunting.” The original table 4 has also been moved to supplementary materials based on feedback from the other reviewer. 

23. Lines 393-399. “Our results emphasize the need ….development potential”. These needs are not directly evident from the results of the present study, rather they appear to reflect the authors’ prior knowledge of these matters. Therefore, these sentences should be appropriately modified, with necessary references if needed.

Thank you for this helpful feedback. We have revised the paragraph and rephrased the statement as follows. “Our results highlight the importance of integrated interventions to prevent undernutrition during first two years of life and observed stunting recovery rates provide promising effects of integrated and targeted nurturing care interventions to avert stunting during infancy and early childhood periods among at risk infants with history of perinatal complications.”

---

## [Editor Report · Decision Letter 1]

13 Mar 2023

Factors associated with recovery from stunting at 24 months of age among infants and young children enrolled in the Pediatric Development Clinic (PDC): a retrospective cohort study in rural Rwanda

PONE-D-22-00485R1

Dear Dr. Nemerimana,

We’re pleased to inform you that your manuscript has been judged scientifically suitable for publication and will be formally accepted for publication once it meets all outstanding technical requirements.

Kind regards,

Gouranga Lal Dasvarma, PhD

Academic Editor

PLOS ONE
---

## [Editor Report · Acceptance letter]

15 Mar 2023

PONE-D-22-00485R1 

Factors associated with recovery from stunting at 24 months of age among infants and young children enrolled in the Pediatric Development Clinic (PDC): a retrospective cohort study in rural Rwanda 

Dear Dr. Nemerimana:

I'm pleased to inform you that your manuscript has been deemed suitable for publication in PLOS ONE. Congratulations! Your manuscript is now with our production department. 

Kind regards, 

on behalf of

Dr. Gouranga Lal Dasvarma 

Academic Editor

PLOS ONE